Natural Hazards and Earth System Sciences (nhess)

# Quantification of Continuous Flood Hazard using Random Forrest Classification and Flood Insurance Claims at Large Spatial Scales: A Pilot Study in Southeast Texas

William Mobley[1], Antonia Sebastian[1,2], Russell Blessing[1], Wesley E. Highfield[1], Laura Stearns[1], and Samuel D. Brody[1]

[1]Department of Marine Sciences, Texas A&M University at Galveston, Galveston, Texas USA
[2]Department of Geological Sciences, The University of North Carolina at Chapel Hill, Chapel Hill, North Carolina USA
**Correspondence:** William Mobley (wmobley@tamu.edu)

**Abstract.** Pre-disaster planning and mitigation necessitates detailed spatial information about flood hazards and their associated risks. In the U.S., the FEMA Special Flood Hazard Area (SFHA) provides important information about areas subject to flooding during the 1% riverine or coastal event. The binary nature of flood hazard maps obscures the distribution of property risk inside of the SFHA and the residual risk outside of the SFHA, which can undermine mitigation efforts. Machine-learning techniques provide an alternative approach to estimating flood hazards across large spatial scales at low computational expense. This study presents a pilot study for the Texas Gulf Coast Region using Random Forest Classification to predict flood probability across a 30,523 ~~km2~~ $km^2$ area. Using a record of National Flood Insurance Program (NFIP) claims dating back to 1976 and high-resolution geospatial data, we generate a continuous flood hazard map for twelve USGS HUC-8 watersheds. Results indicate that the Random Forest model predicts flooding with a high sensitivity (AUC 0.895), especially compared to the existing FEMA regulatory floodplain. Our model identifies 649,000 structures with at least a 1% annual chance of flooding, roughly three times more than are currently identified by FEMA as flood prone.

## 1 Introduction

In the US, pluvial and fluvial flood events are some of the most damaging environmental hazards, averaging $3.7 billion annually, totaling over $1.5 trillion in total losses since 1980 (for Environmental Information , NCEI). This trend represents an increase of about 15% in flood losses per year since 2002, despite large scale efforts to mitigate losses over the same period (Kousky and Shabman, 2017). To offset the rising costs associated with extreme flood events, pre-disaster planning and mitigation necessitates detailed spatial information about flood hazards and their associated risks.

In the US, the Federal Emergency Management Agency (FEMA) Special Flood Hazard Area (SFHA) – the area of inundation associated with a 1% annual exceedance probability – provides a basis for community and household planning and mitigation decisions (Blessing et al., 2017). These maps, intended to be used to set flood insurance rates, have become the de facto indicator of flood risk nationwide and are the primary reference point when making a vast array of decisions related to flood

risk such as where it is safe to develop, household protective actions, and local mitigation policies. However, because the SFHA is binary, little information is provided about the distribution of risk to properties inside of the mapped flood hazard areas, and,

even more so, for residual risks to properties outside of the mapped flood hazard areas. Thus, the floodplain boundaries result in "dichotomous decisions" whereby a house is treated the same whether it is 1 meter or 1 kilometer outside of the floodplain boundary (Brody et al., 2018; Morss et al., 2005). As a result, homes are being built and purchased in at-risk areas, and actions that would increase flood resilience, such as meeting ~~National Flood Insurance Program (NFIP )~~ NFIP minimum design and construction requirements, are not being adopted.

Further compounding the SFHA's inability to indicate at-risk areas is that many of the nation's flood hazard maps are out of date, or ~~,~~ worse, non-existent. For example, a 2017 study found that over a quarter of high risk counties have U.S. flood hazard maps over 10 years old, failing to capture recent changes in climatology and land use/land cover that can heighten risk (Office, 2017). Although FEMA's Map Modernization (Map Mod) program updated many of the nation's flood maps from 2003 to 2008 it still struggles to keep them up to date (ASFPM, 2020) in part because the models used to produce the FEMA SFHAs

are discontinuous across large spatial scales and often commissioned in patchwork, community-by-community basis that is both slow to implement and resource intensive ((FEMA) 2015). A recent study found that 15% of the NFIP communities have flood hazard maps that are over 15 years old, and that only a third of that nation's streams have flood maps (ASFPM, 2020). Updating the nations floodplain maps and mapping previous unmapped areas is a costly solution at an estimated $3.2 billion to $11.8 billion (USD) with an additional $107 million to $480 million (USD) per year to maintain them (ASFPM, 2020).

An alternative approach that is gaining popularity are models that can leverage both big data (e.g., large-scale remotely sensed imagery) and high-performance computing environments to efficiently estimate flood hazard at continental (Wing et al., 2018; Bates and global scales (Ward et al., 2015; Ikeuchi et al., 2017). These models overcome some of the limitations of community-by-community watershed-scale modeling by providing a methodologically consistent approach using comprehensive spatial datasets that is less resource intensive. Although capable of accurately identifying general patterns of flood hazard over large areas, these

models often struggle to correctly estimate small scale variability in stormwater dynamics within highly developed areas. That is, model performance at community and household scales is highly dependent on detailed information about local-scale hydrology and flood control infrastructure, with limited observational data to validate the results. Moreover, these models still suffer from the same sources of uncertainty found at the watershed-scale including, specifically, natural uncertainty arising from physical processes driving runoff (Singh, 1997), uncertainty found within model parameterization (Moradkhani and Sorooshian, 2009)

, and uncertainty arising from the quality or availability of input data (Rajib et al., 2020). Accounting for uncertainty at locally-relevant scales would likely require significant increases in observational hydrometeorological data and immense financial resources.

Machine learning (ML) methods provide a potential alternative to estimate flood hazards based on historical records of flood loss, especially in resource-limited areas. One such ML algorithm that has shown to be particularly effective within flood hazard

mapping are random forests which have recently been used to create a spatially complete floodplain map of the conterminous United States (Woznicki et al., 2019) and to predict flood insurance claims at US Census tract and parcel levels (Knighton et al., 2020). Although initial work has shown random forests improve the prediction of flood hazard, there have been no studies that

have used historic records of structural flood damage to estimate a probabilistic floodplain~~, and many of the previous studies have used sampling procedures that can undermine model reliability~~. Also, there has been little to no effort to compare random forest predictions against existing regulatory floodplains.

We address these gaps by introducing a novel method to map flood hazards continuously across large spatial scales using a random forest classification procedure trained on 40 years of historic flood damage records from the NFIP. Using the NFIP data and high-resolution geospatial data (e.g., topographic, land use/land cover, soils data), we generate flood hazard maps for a large coastal area in Texas Gulf Coast Region. We then compare our modeled outputs against the FEMA floodplains using multiple metrics at regional and community scales. The following sections provide further background on machine learning algorithms and their application to hazards research (Section 2), describe the methods and data used for our analysis (Section 3) and present the model results (Section 4). This is followed by a discussion (Section 5) and conclusions (Section 6).

## 2 Background

Data-driven models – those that use statistical or machine learning algorithms for empirical estimations – are prevalent in water resources research (Solomatine and Ostfeld, 2008) and are rapidly gaining popularity for prediction and estimation of flooding in the hydrological sciences (Mosavi et al., 2018). For instance, models predict stream discharge rates (Albers and Déry, 2015), estimate insured flood damage for either the household (Wagenaar et al., 2017) or aggregated (Brody et al., 2009). Data driven approaches have also identified flood probability during a flood event (Mobley et al., 2019)(i.e. flood hazard). When data in an area is sparse these models can help better describe the system (Rahmati and Pourghasemi, 2017). Often however, these approaches require large datasets for an accurate representation (Solomatine and Ostfeld, 2008). In answer to the data problem many data driven models rely on non-traditional data sources. For example, using video frames flood waters can be identified for a given location (Moy de Vitry et al., 2019).

Flood hazard models use dichotomous variables and driver layers to predict the likelihood of flooding across the landscape. This dichotomous variable can come from a variety of datasets, such as high water marks or flood losses (Knighton et al., 2020). Depending on data availability, models have predicted flood hazard across time (Darabi et al., 2019) or for specific events (Lee et al. 2017). Producing one of two outputs a probability of flooding (Darabi et al., 2019) , or refined into classes of susceptibility (Darabi et al., 2019; Dodangeh et al., 2020). Both of which give more information than the current SFHA dichotomy, in or out of the flood zone. ~~Through~~ By modeling flood hazard through this process, researchers are able to map large geographic areas (Hosseini et al., 2019) with potential to scale further given sufficient data availability. This method has shown to quickly (Mobley et al., 2019) and accurately estimate flooding (Bui et al., 2019) .

The amount of data available and how its structured affect the techniques available for predicting flood hazard. ~~Flood~~ Data-driven flood hazard models require point locations where flooding has occurred. If non-flooded locations are unavailable pseudo-absences can be randomly generated for modelling (Barbet-Massin et al., 2012). Common algorithms used for predicting flood hazard models include neural networks (Janizadeh et al., 2019a; Bui et al., 2019), Support Vector Machines (SVM) (Tehrany et al., 2019a), and Decision Trees, such as Random Forests (Woznicki et al., 2019; Muñoz et al., 2018). While

other algorithms have been used for predicting flood hazard (Bui et al., 2019), these three algorithms are often used in machine learning due to their maturity in research and their generalizability. SVMs are ideal in areas with small sample sizes, but computation times are quadratic as sample sizes increase (Li et al., 2009). The computational complexity of the model removes the scalability of the model, a primary benefit over physical models. Neural networks are often cited as highly accurate (Mosavi et al., 2018), but often come with a reproducibility problem (Hutson, 2018) .

Finally, computational requirements for decision trees are lower than the other two algorithms. The Random Forest algorithm comes from the decision tree family of models. The Random Forest model is highly generalizable within the drivers' parameters. Decision trees are non-parametric and use logic to branch at different values within the independent variables to best fit the classification (Quinlan 1986). By creating numerous trees and democratizing the decision, ensemble classifiers reduce overfitting of the final model while maintaining accuracy (Breiman 2001). Each tree is given a random subsample of the independent variables to predict the dependent variable. These ensemble classifiers are computationally efficient and maintain a high degree of sensitivity (Belgiu and Drăgut, 2016). Random forests are capable of identifying interactions between independent variables and the dependent variable regardless of the effect size (Upstill-Goddard et al., 2013).

## 3 Methods and Materials

### 3.1 Study Area

We use the USGS Watershed Boundary Dataset to delineate a region encompassing thirteen Hydrologic Unit Code (HUC) 8-digit watersheds that drain to Galveston Bay and the intercoastal waterway, as well as the Lower Trinity and Brazos Rivers and the San Bernard River (Figure 1). The resulting study area encompasses 28,000 km$^2$ and includes two economic population centers: the Houston-Woodlands-Sugar Land Area, also known as Greater Houston, home to 4.8 million people and the Beaumont-Port Arthur-Orange Area, also known as the Golden Triangle, home to 0.4 million people (Bureau 2019). Including rural areas, the total population of the study region is estimated to be around 8 million, accounting for around 25% of the population of Texas (Bureau, 2019).

The region is prone to damaging flood events, resulting in $16.8 billion in insured loss between 1976 and 2017 across 184,826 insurace claims. Predominately clay soils and low topographic relief coupled with extreme precipitation results in wide and shallow floodplains. Regional flood events are driven by several dominant mechanisms, including mesoscale convective systems (MCS) and tropical cyclones (Van Oldenborgh et al., 2017). Recent examples of MCS-driven events include the Memorial Day Flood (2015), Tax Day and Louisiana-Texas Floods (2016). Several stalled tropical cyclones, including Tropical Storms Claudette (1979), Allison (2001), and Hurricane Harvey (2017), have also resulted in record-setting precipitation. Frequency estimates of tropical cyclone landfall range from once every nine years in the eastern part of the study region to once every nineteen years in the southwest part of the region (NHC 2015). Historical storm surge ranges as high as 6 m along the coast. Notable surge events include the Galveston Hurricanes (1900, 1915), Hurricane Carla (1961), and Hurricane Ike (2008). Several previous studies have demonstrated that flood hazards are not well-represented by the FEMA SFHA (Highfield

et al., 2013; Brody et al., 2013; Blessing et al., 2017), suggesting a large proportion of the population along the Texas coast are not only vulnerable to flooding, but the decision makers lack the information to properly account for it.

## 3.2 Data Collection

### 3.2.1 NFIP Claims

~~Between~~ NFIP flood claims were used as the predictive variable. NFIP claims between 1976 and 2017 ~~, there were~~ were available for the study totaling 184,826 ~~NFIP~~ claims, each of which was geocoded to the parcel centroid. The NFIP flood losses dataset provides the location, total payout, and structural characteristics. Claims provide a reliable indication of the presence of flooding but fail to identify locations that have not flooded making the dataset presence only. Random Forest algorithms require a binary dependent variable identifying presence and absence locations. A pseudo-random sample of background values can be used as a proxy for locations where flooding is absent (Barbet-Massin et al., 2012). When used in this way, the pseudo-absences essentially represent the population that could potentially flood. Therefore, the background sample locations were based on a random selection of ~~structures~~ all structures within the study area. The Microsoft structures footprints dataset was used due to the open source availability and high accuracy (https://github.com/Microsoft/USBuildingFootprints). The study matched structures and claims using a one-to-one sample by watershed and year. Meaning that for each claim, a structure is selected given the same year and located within the same watershed. This one-to-one matching reduces potential bias from an unbalanced dataset (Chen et al., 2004). The final dataset removed any sample where an independent variable had null values, but the final ratio remained close to 50% of the claims with a sample size of 367,480.

### 3.2.2 Contextual Variables

To parameterize flood hazard, several contextual variables were considered which represent the potential predictors of flooding across the study area (see Table 1). These variables can be divided into two main categories: (1) topographic: elevation and distance features which drive watershed response; and (2) hydrologic: overland and soil characteristics which govern infiltration and runoff. The variables were collected at different scales based on data availability. All variables were resampled to a 10-meter raster and snapped to the Height Above Nearest Drainage (HAND) dataset. Below, we outline the variables, the reasoning behind their inclusion, and previous data-driven flood hazard studies that used them.

### 3.2.3 Hydrologic

Hydrologic variables explain how stormwater moves across the landscape and, therefore, can help differentiate between low and high flood potential areas. The amount of stormwater that a given area can receive is a function of its flow accumulation potential, which is primarily mediated by three factors: the soil's ability to absorb water (i.e. saturated hydraulic conductivity), roughness (i.e. Manning's *n*), and imperviousness. Flow accumulation measures the total upstream area that flows into every raster cell based on a flow direction network as determined by the NED (Jenson and Domingue, 1988). Areas with high flow accumulation are more susceptible to receiving larger amounts of stormwater during a given rainfall event. Soil infiltration

influences the speed and amount at which stormwater can be absorbed into the ground. When stormwater cannot move into the ground easily, it may result in additional runoff, particularly in urbanized and downstream areas. Two measures of soil infiltration include lithology and saturated hydraulic conductivity (Ksat), both of which have been shown to be strong predictors of flood susceptibility (Brody et al., 2015; Janizadeh et al., 2019b; Hosseini et al., 2019; Mobley et al., 2019). For this study, Ksat values were assigned to soil classes obtained from the Natural Resources Conservation Service's (NRCS) Soil Service Geographic Database (SSURGO) using the values presented in Rawls et al. (1983), and then averaged across the upstream contributing area for each cell.

Imperviousness is another strong indicator of an area's ability to infiltrate water. Previous studies have found that increasing impervious surfaces as a result of urbanization reduces infiltration and causes increased surface runoff and larger peak discharges making it an important aspect in determining flood frequency and severity (Anderson, 1970; Hall, 1984; Arnold Jr and Gibbons, 1996; White and Greer, 2006). Imperviousness has been previously shown to be highly important in the Houston region where urban sprawl has greatly increased imperviousness in the region and contributed to higher volumes of overland runoff (Brody et al., 2015; Gori et al., 2019; Sebastian et al., 2019). For this study, percent impervious was measured using the percent developed impervious surface raster from the National Land Cover Database (NLCD) for the years: 2001, 2006, 2011, and 2016. Values range from 0-100% and represent the proportion of urban impervious surface within each 30-m cell. Because comparable remote sensing imagery only exists for these year, impervious surface was associated with the nearest date of each claim.

Roughness influences the speed at which stormwater can move across the landscape as well as the magnitude of peak flows in channels (Acrement and Schneider, 1984). Previous engineering studies have corroborated the relationship between roughness and overland water flow using Manning's roughness coefficient (Anderson et al., 2006; Thomas and Nisbet, 2007). This coefficient, called Manning's n, has become a critical input in many hydrological models and has also been shown be a good predictor of event-based flood susceptibility (Mobley et al., 2019). For this study, roughness values were assigned to each NLCD land cover class using the values suggested by Kalyanapu et al. (2010), and, like Ksat, was averaged across the contributing upstream area for each raster cell for the years: 2001, 2004, 2006, 2008, 2011, 2013, and 2016.

### 3.2.4 Topographic

Elevation and slope are topographic variables frequently used to model flood hazard (Lee et al., 2017b; Tehrany et al., 2019a; Rahmati and Pourghasemi, 2017; Bui et al., 2019; Hosseini et al., 2019; Mobley et al., 2019; Darabi et al., 2019). Low-lying areas tend to serve as natural drainage pathways making them more susceptible to flooding and ponding during rainfall events. Elevation and slope were calculated using the National Elevation Dataset (NED), which was provided as a seamless raster product via the LANDFIRE website at a 30-m resolution LandFire (2010).

Three continuous proximity rasters were used in this study: distance to stream, distance to coast, and height above nearest drainage (HAND). Proximity to streams and the coastline have been shown to be significant indicators of flood damage Brody et al. (2015) as these areas are typically much more prone to overbanking and surge respectively. More recent flood hazard studies have used proximity to streams (Lee et al., 2017b, a; Dodangeh et al., 2020; Janizadeh et al., 2019a, b; Tehrany et al.,

2019a, b; Rahmati and Pourghasemi, 2017; Bui et al., 2019; Hosseini et al., 2019; Mobley et al., 2019; Darabi et al., 2019), whereas proximity to coasts has been less common (Mobley et al., 2019). Both distance to stream and coast were calculated based on the National Hydrography Dataset (NHD) stream and coastline features.

HAND is calculated by defining the height of a location above the nearest stream to which the drainage from that land surface flows (Garousi-Nejad et al., 2019). Areas with high measures of HAND are more buffered from flooding because it requires increasingly more stormwater of short durations to create the peak flows that would reach those locations. This measure has been used to calculate flood depths, the probability of insured losses from floods (Rodda, 2005), soil water potential (Nobre et al., 2011), groundwater potential (Rahmati and Pourghasemi, 2017), and flood potential (Nobre et al., 2011). HAND was downloaded from the University of Texas' National Flood Interoperability Experiment (NFIE) continental flood inundation mapping system (Liu et al., 2016) at a 10-m resolution.

Topographic wetness index (TWI) (Beven and Kirkby, 1979) is a popular measure of the spatial distribution of wetness conditions and is frequently used to identify wetlands. TWI is used to quantify the effects of topography on hydrologic processes and is highly correlated with ground water depth, and soil moisture (Sörensen et al., 2006). This measure has been found to be an influential and, in some cases, a significant predictor for estimating flood hazard (Lee et al., 2017b; Tehrany et al., 2019a; Bui et al., 2019; Hosseini et al., 2019; Tehrany et al., 2019b). TWI is calculated by the following equation:

$$TWI = ln\frac{(A*900)+1}{tan(\frac{(S_0*\pi)}{180})} \tag{1}$$

Where A is the contributing area (or flow accumulation) and $S_0$ is the average slope over the contributing area. High values of TWI are associated with areas that are concave, low gradient areas where water often accumulates and pools making them more vulnerable to flooding. ~~Random Forest Model Random Forest~~

### 3.3 **Random Forest Model**

Random Forest models categorize samples based on the highest predicted probability for each class. We developed a random forest algorithm at the parcel level ~~using the variables in Table 1.~~ (Figure 2). The NFIP flood claims dataset was split between a training and test dataset. The test dataset was based on 30% of the initial dataset. Within the training dataset cross-validation is used to decide the final variables, and parameters. A k-folds sampling method uses 90% of the training dataset to predict the other 10%. This sampling method helps to measure the robustness of the model, as variables are pruned, and parameters are tweaked. The cross-validation assessment uses a 10-sample stratified k-fold. Structures were randomly sampled reducing the potential for storm event-based bias. In addition to the k-folds cross-validation, a year-by-year assessment was performed by removing one year as a validation sample for a model based on all other years. For example, 1976 through 2016 were used to predict flood hazard in 2017, then 1977 through 2017 were used to predict 1976, and so forth. This year-by-year assessment helps to identify the limitations of this flood hazard approach. All random forest computations were performed with the sci-kit learn package (Pedregosa et al., 2011) in Python version 3.8.

Creating a properly calibrated Random Forest requires tuning two parameters to minimize error, and variable selection to improve generalizability. The two parameters tuned were the number of trees and the maximum tree depth. The number of trees controls how large a forest is used for the predictive model. Increasing the number of trees reduces error rates and increases the attributes used in the decision (Liaw et al., 2002), but comes at the costs of increasing computation time. Tree depth controls the maximum number of decisions that can be made, but too large of a tree will increase the chance of the model overfitting the data and reduce generalizability. The model used 200 trees, and a maximum tree depth of 90, after optimizing error rates using the k-folds analysis. Variable reduction reduces the complexity of the model and decreases the likelihood of the model overfitting, while speeding up the final training and raster predictions. An out-of-bag error score (OOB), isolates a subset of the training dataset which is used to measure error rates of each variable (Breiman, 1996) and generates feature importance. Initially, all variables were added to the model, those variables with the lowest contribution to feature importance were removed from the final model. ~~A figure representing feature importance can be found in Appendix A~~Two variables were removed from the final model for not meeting the required threshold TWI and Flow Accumulation (Figure 3).

A series of metrics were used to identify if the model is properly calibrated. Average accuracy and sensitivity are measured for each iteration of the k-fold analysis. Accuracy measures the percent of correctly identified flooded and non-flooded samples. Sensitivity estimates the probability that flooded sample will be predicted with a higher likelihood of flooding than a non-flooded sample (Metz, 1978) and based on the Area Under the Curve (AUC) of the Receiver Operating Characteristic (ROC). In practice AUCs are robust for continuous probabilities and are not limited by the number of features in a model or a threshold to use for classification. Both accuracy and sensitivity were compared against the final model and the test sample. Ideally, the final model should be similar in accuracy and sensitivity to the average found in the k-fold analysis. A calibration plot shows how well the model predicts probabilities given the proportion of flooded points in each bin. A properly configured model will fall along a diagonal on the plot. Random forest outputs can be either a classification or represented as a probability. In this study, the final output was based on probability of flooding.

As an additional validity check we compared the RF prediction against the FEMA SFHAs by examining the amount of area and structures exposed to specific annual exceedance probabilities. The RF model predicts the probability that a location floods over the 42 years of the study. To convert the RF probabilities to annual exceedance probabilities we used the following equation.

$$exceedence = 1 - (1 - p_{flood})^{(1/T)} \tag{2}$$

This procedure allows for a more direct comparison between the FEMA SFHA and flood hazard.

## 4 Results

### 4.1 Model Performance

Ten k-fold cross validation was performed to predict flood hazard across twelve watersheds in the study area. From the k-folds analysis, the mean model accuracy was $81.9\% \pm 0.00198$, while the final test accuracy is $82.2\%$. The model had an average

sensitivity ~~was~~ of 0.893 $\pm$ 0.00184 (Figure ~~2~~ 4 left) while the final model produced a sensitivity of 0.895. In the year-by-year analysis, years predicted with a high sensitivity fall within relatively normal events, while the extreme events such as hurricane Harvey (60+ inches of rain) and Hurricane Ike (high storm surge) ~~should~~ perform poorer in relation. The year-by-year analysis showed more variation in sensitivity (Figure ~~2~~ 4 right), mean AUC is 0.884$\pm$0.0482. The year with Hurricane Ike had the lowest sensitivity at 0.730 and the year with Hurricane Harvey performed poorly as well 0.761. The highest sensitivity was predicted in 1997: 0.929. This analysis sheds light on how the model performs for extreme events. For example, years such as 2017 perform worse as the conditions brought on by Hurricane Harvey are rarely seen.

The calibration plot (Figure ~~3~~ 5) suggests that the model slightly underpredicts flooded points at lower probabilities, and slightly overpredicts flooded points at higher probabilities. The underprediction is explained by the histogram below the plot. Most non-flooded structures have a probability below 30%, while the largest proportion of flood loss points occur above 90%.

## 4.2 Comparison with the FEMA SFHA

Flood hazard probabilities were converted to annual exceedance (Figure ~~4~~ 6) which allowed us to compare the amount of area and structures exposed to specific annual exceedance probabilities with the FEMA SFHAs. Note as a reminder, the 100-year floodplain has an annual exceedance of 1% and the 500-year floodplain has an annual exceedance of 0.2%. Based on the modeled flood hazard, 13,810 $km\hat{2}$ of land and 649,140 structures have a 1% chance of flood any given year and 26,348 ~~km2~~ $km^2$ and 1.81 million structures have 0.2% chance of flooding any given year. In contrast, 8,000 ~~km2~~ $km^2$ are classified within the 1% SFHA encompassing 207,000 structures, while the 0.2% floodplain increases in size to 9,900 ~~km2~~ $km^2$ and encompasses 500,000 structures.

Focusing on three flood prone areas, the model shows high flood hazards currently not identified by the FEMA SFHA (Figure ~~5~~ 7). In Conroe (Figure ~~5a-c~~ 7a-c), for example, the 1% areas appear visually similar for the SFHA and the flood hazard model, but when counted the flood hazard model predicts seven and a half times more structures around lake Conroe that have at least a 1% chance annually of flooding compared with the 1% SFHA. In the Meyerland area (Figure ~~5d-f~~ 67d-f) both the FEMA SFHA and the Flood Hazard model clearly predict flooding along the rivers and identify areas where the floodplain expands. However, the flood hazard model identifies a much larger area with at least a 1% chance of annualized flooding. Finally, within Port Arthur (Figure ~~5g-i~~ 7g-i) the flood hazard model expects that the whole area has a flood hazard of at least 0.2%, a significantly larger area than FEMA's 0.2% floodplain.

## 5 Discussion

The results illustrate that flood hazards can be accurately estimated using a machine learning algorithm. The model is computationally efficient, scalable, and can be used to predict flood hazards over relatively large regions. ~~Overall, the model creates an accurate representation of flood hazard for~~ Training and image prediction was run in 1-hr on a high end desktop computer, in comparison physical models can run for 20 minutes or up to 10 hours for an area 0.15% of the study area ~~and demonstrates strong discriminatory power~~ (Apel et al., 2009). Overall, the model demonstrates strong predictive power for estimating historic

damages and accurately represents the spatial distribution of the flood hazard across those 40-years of flood damages. When compared against similar studies using machine learning approaches to predict flood hazards (Dodangeh et al., 2020; Janizadeh

et al., 2019a), our model demonstrates lower sensitivity. The differences are likely attributed to the high topographic relief of these regions where fluvial flood hazard predominate (Tehrany et al., 2019b), likely contributing to the predictive capacity of the models. In contrast, Southeast Texas is characterized by little topographic relief where flooding may emanate from pluvial, fluvial, and marine sources, making flood prediction more complex. In fact, when comparing model sensitivities across the study region, we find that the model performance increases in more steeply sloped inland areas than flat coastal areas.

Another aspect impacting model performance was sample selection. A systematic approach that identifies areas that did not flood (Darabi et al., 2019) can be important to increasing model performance (Barbet-Massin et al., 2012). While the model is based on a comprehensive record of observed flood claims in the study area from 1976 to 2017, there is no comparable record for structures that have not flooded. One option would be to randomly sample areas that have no claims, however this would not control for bias in the absence data and come at the expense of model performance (Wisz and Guisan, 2009). To overcome this

potential bias, we generated pseudo-absences by randomly selecting a sample of non-flooded structures by watershed and year to minimize this selection bias (Phillips et al., 2009). Based on the calibration plot (shown in Figure 35) this is an appropriate assumption.

Another source of bias comes from how contextual variables were associated with the claims dataset. That is, the contextual variables that were attached to the claims came from the closest year for which the data was made available. However, this

can cause some skewed behavior, particularly for the older claims, because some of the contextual variables only start in 2001 whereas the claims date back to 1976. Variables impacted the most by this are the result of large-scale changes in urbanization over time, which include imperviousness and roughness. More specifically, the imperviousness and roughness conditions attached to claims is going be less representative of the actual conditions during the time of loss the further one goes back in time. However, a large majority of the claims (approximately 75%) occurred after 2001 and most of the contextual

variables do not change, or change minimally, over time (e.g. HAND, elevation, distance to coast, etc.) mitigating the influence of this bias.

A novel outcome of this analysis are statistically generated flood hazard maps that can be compared to FEMA's regulatory floodplains. That is, we used the predicted model likelihoods to generate 1% and 0.2% annual exceedance probability thresholds, or ,equivalently, the 100-year and 500-year flood hazard areas. The statistically generated flood hazard areas differed from

the regulatory floodplains in that they are: 1) nearly 3 times as large and 2) captured areas that are hydrologically disconnected from streams and waterbodies. The findings suggest that this approach can better capture small scale variability in flood hazard by implicitly detecting underlying drivers that manifest themselves through subtle changes in historic damage patterns and trends. This corroborates previous research findings that the current 100-year floodplain underestimates and fails to accurately represent flood risk particularly in urban areas (Blessing et al., 2017; Galloway et al., 2018; Highfield et al., 2013).

It should be noted that the interpretation of the predicted random forest flood hazard areas differs from existing regulatory floodplains, in that they are detecting the return period for structurally-damaging inundation. The differences between the statistically generated flood hazard areas and regulatory floodplains is likely a result of multiple advantages of a data-driven

approach in identifying the conditions in the underlying drivers (Elith et al., 2011). In other words, a data-driven model to better capture the reality of flood hazard by using actual historic impacts and simultaneously identifying small-scale variations in flood exposure. By using historic losses, the random forest model accounts for extreme events that have occurred, but projecting future hazard is currently limited. The model does not currently incorporate precipitation patterns, ~~this was by design as we assumed that with the relatively small-scale, precipitation would be homogenous around the region and add little insight~~. Future work should examine the sensitivity of the model to precipitation as an input. With precipitation added to the model, there is still a potential for error by underestimating the probability of future extreme events.

## 6  Conclusions

In this paper we demonstrate the efficacy of a random forest statistical model in spatially identify flood hazards in Southeast Texas, encompassing the Houston metropolitan area. In comparison with FEMA SFHA, we show that a statistical machine learning flood hazard model can 1) better capture the reality of flood hazard by using actual historic impacts, 2) better capture small scale variability in flood hazard by implicitly detecting underlying drivers that manifest themselves through subtle changes in historic damage patterns and trends, 3) avoids the uncertainty associated with estimating rainfall return periods, stormwater infrastructure characteristics, and flood depths, 4) easily include alternative drivers of flood hazard such as HAND and ~~TWI, and~~ 5) be quickly updated using recent insurance claim payouts.

*Data availability.*  All independent drivers for the flood hazard model can be found at the at the Dataverse repository (https://dataverse.tdl.org/dataverse/M3FR).The Flood hazard output can be found at the following url: https://doi.org/10.18738/T8/FVJFSW (Mobley, 2020). Flood loss data can not be publicly shared due to privacy concerns. The sources of python libraries used are as follows: Scikit-learn library (Pedregosa et al., 2011); RasterIO (Gillies et al., 2013–); NumPy (van der Walt et al., 2011); and Pandas (Wes McKinney, 2010).

*Author contributions.*  WM, RB, AS, WH, and SB designed the analysis. LS curated data. WM developed the model code and performed analysis. AS and RB performed model validation. LS created Visualizations. WM prepared the manuscript with contributions from all co-authors. SB and WH acquired funding.

*Competing interests.*  The authors declare that they have no conflict of interest.

*Acknowledgements.*  This research was supported by the NSF PIRE Grant No. OISE-1545837 and funding from the Texas General Land Office Contract No. 19-181-000-B574.

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

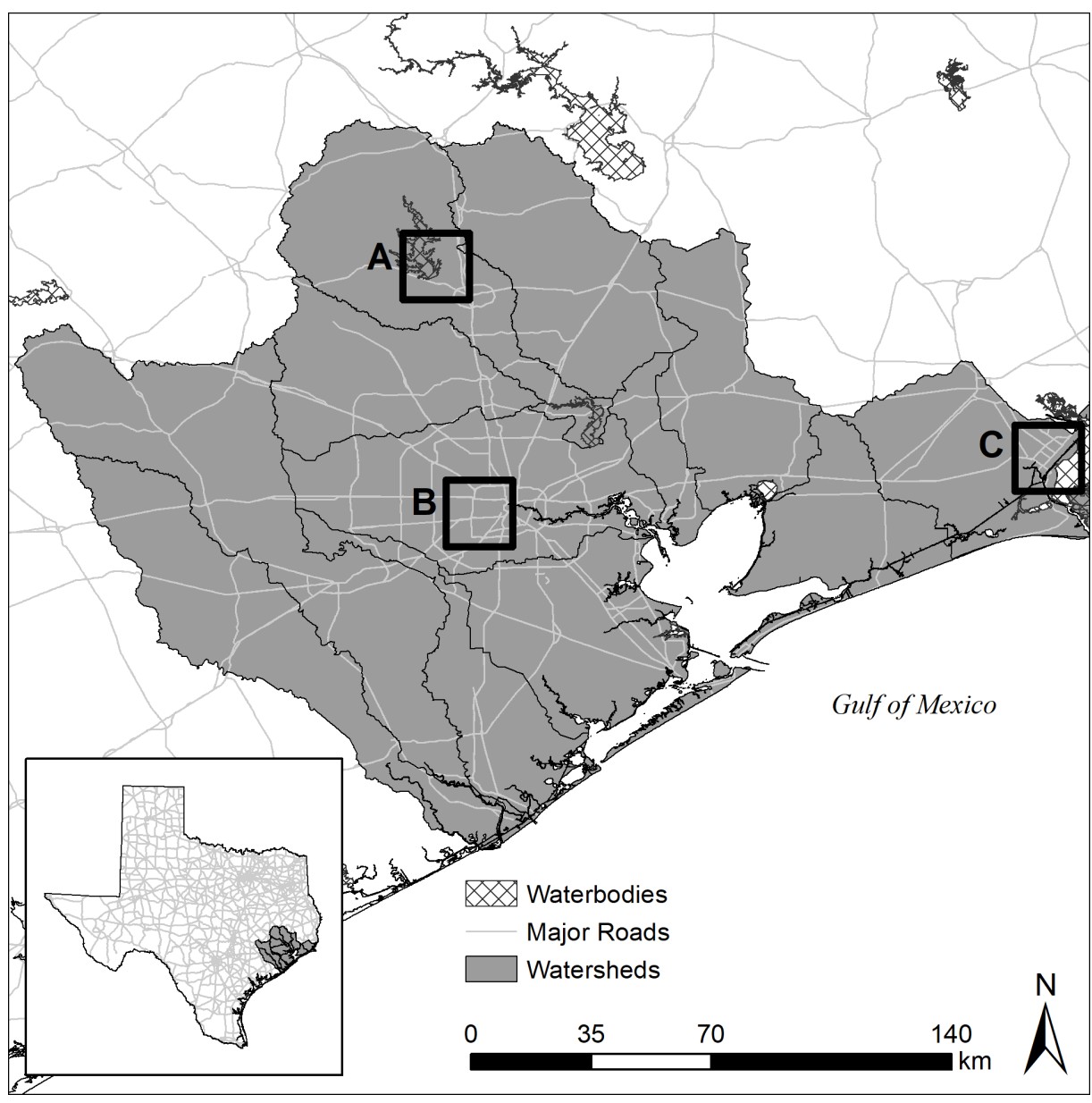

**Figure 1.** Depicts a map of the Study Area. Data originated from the U.S. Census Bureau and the U.S.Geological Survey.

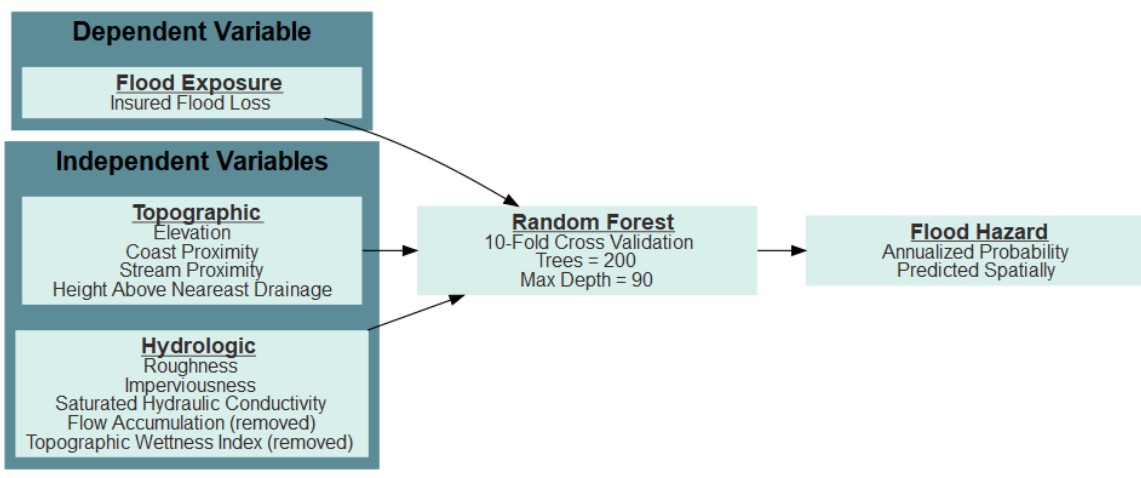

**Figure 2.** Conceptual frame work to develop the annualized flood probability.

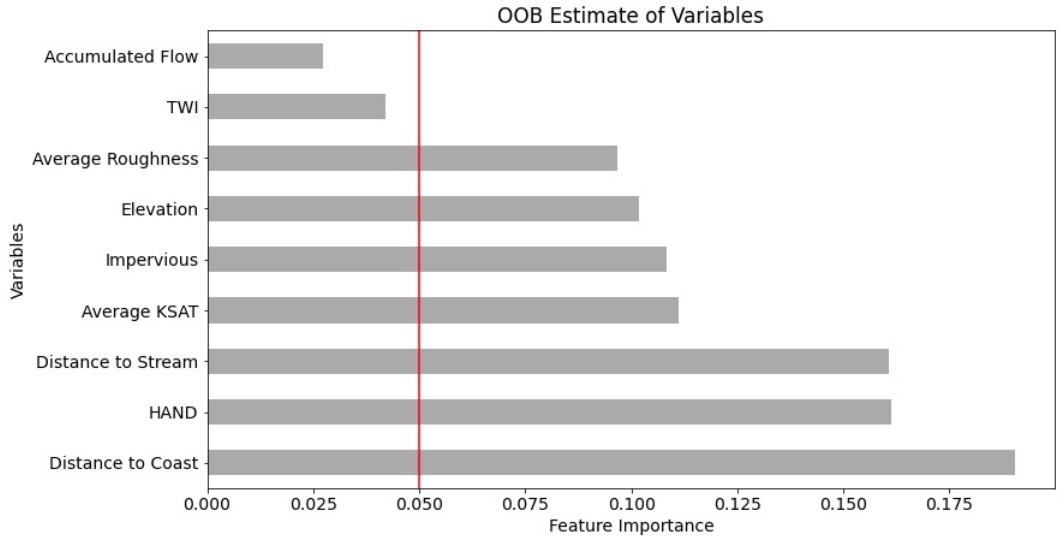

**Figure 3.** Feature importance chart. Note variables below, 0.05 were removed from the final model. The red line denotes the cutoff value for the final model.

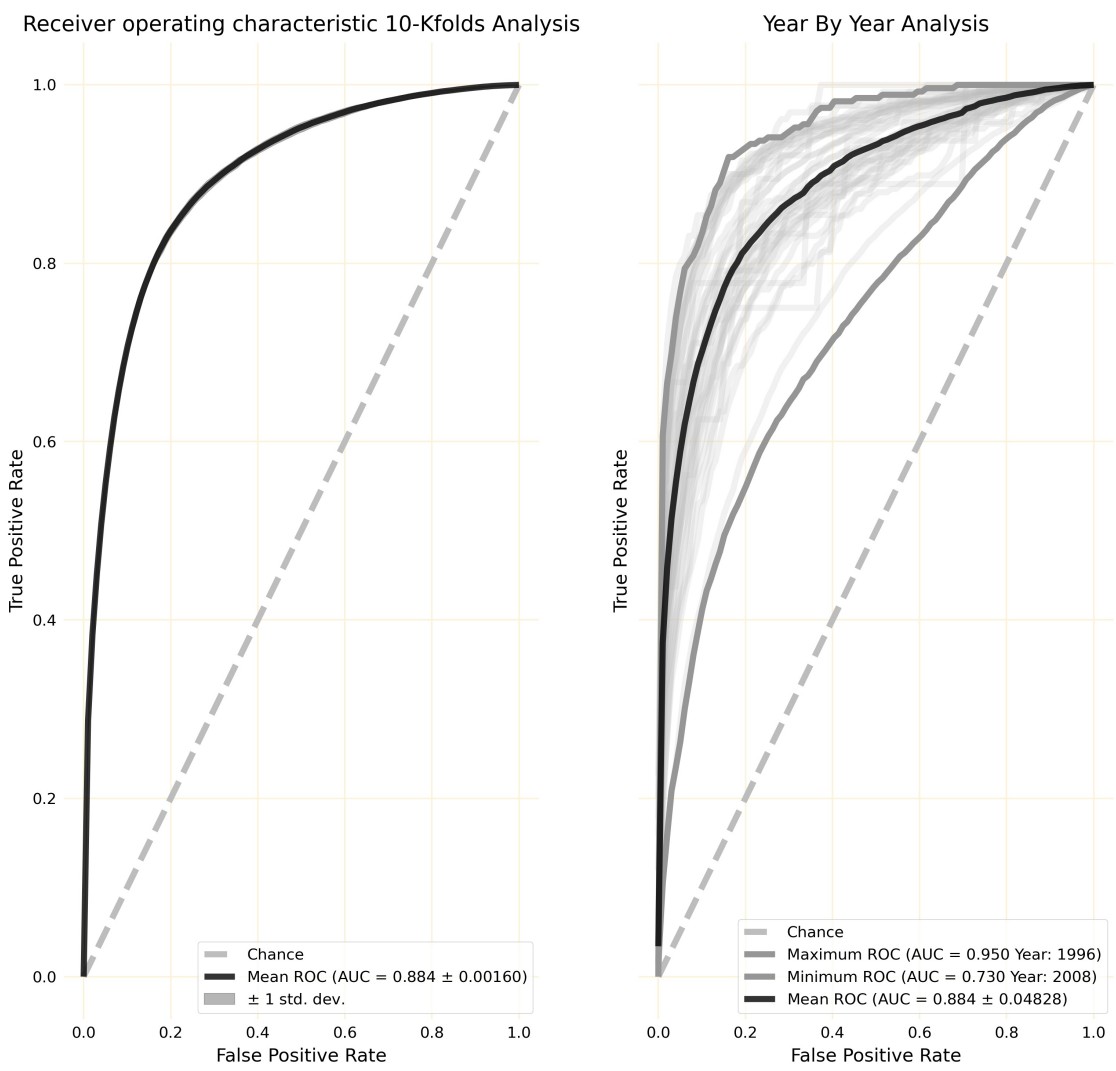

**Figure 4.** Left) The Receiver Operating Characteristic (ROC) shown for 10k-folds cross validation. Right) The ROC shown for the year-by-year analysis. The year by year analysis shows generalizability of the model by predicting how well a standalone years flooding will fit within the model. This approach also identifies years with storm outliers.

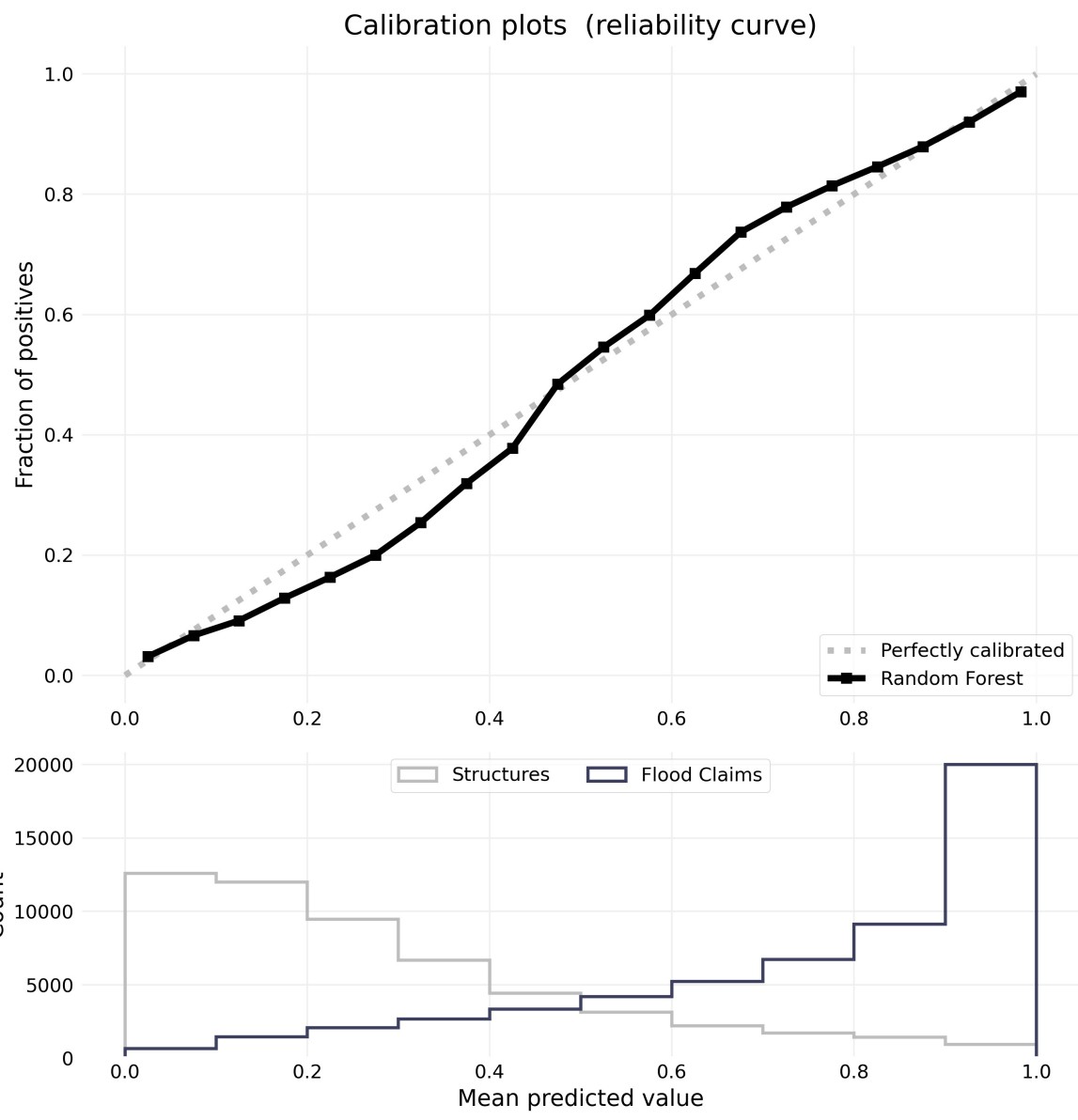

**Figure 5.** Calibration plot (top) with histogram of flooded / non-flooded samples and their predicted values.

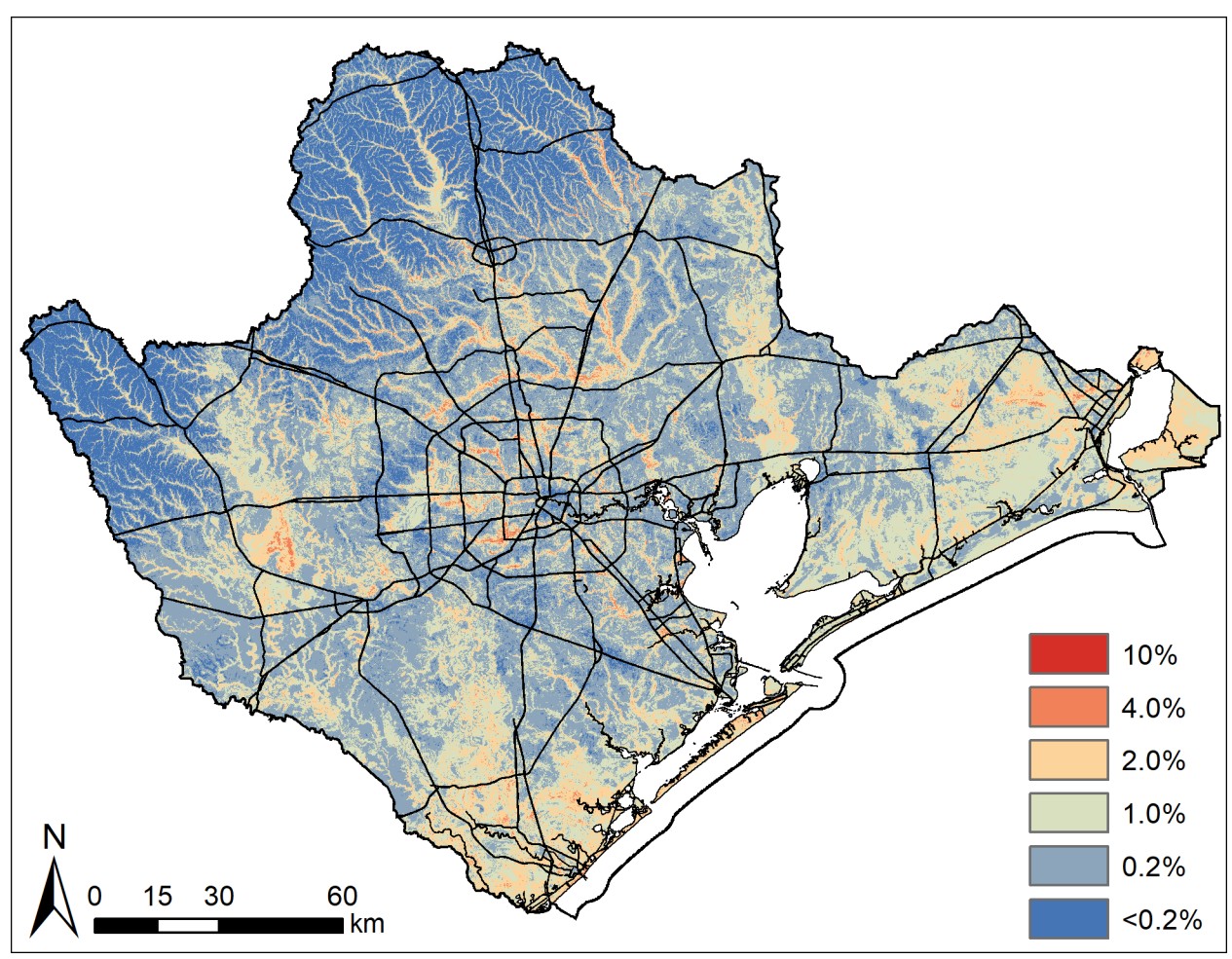

**Figure 6.** Continuous flood hazard map for the pilot study area based on model output. Road layers originated from the U.S. Census Bureau.

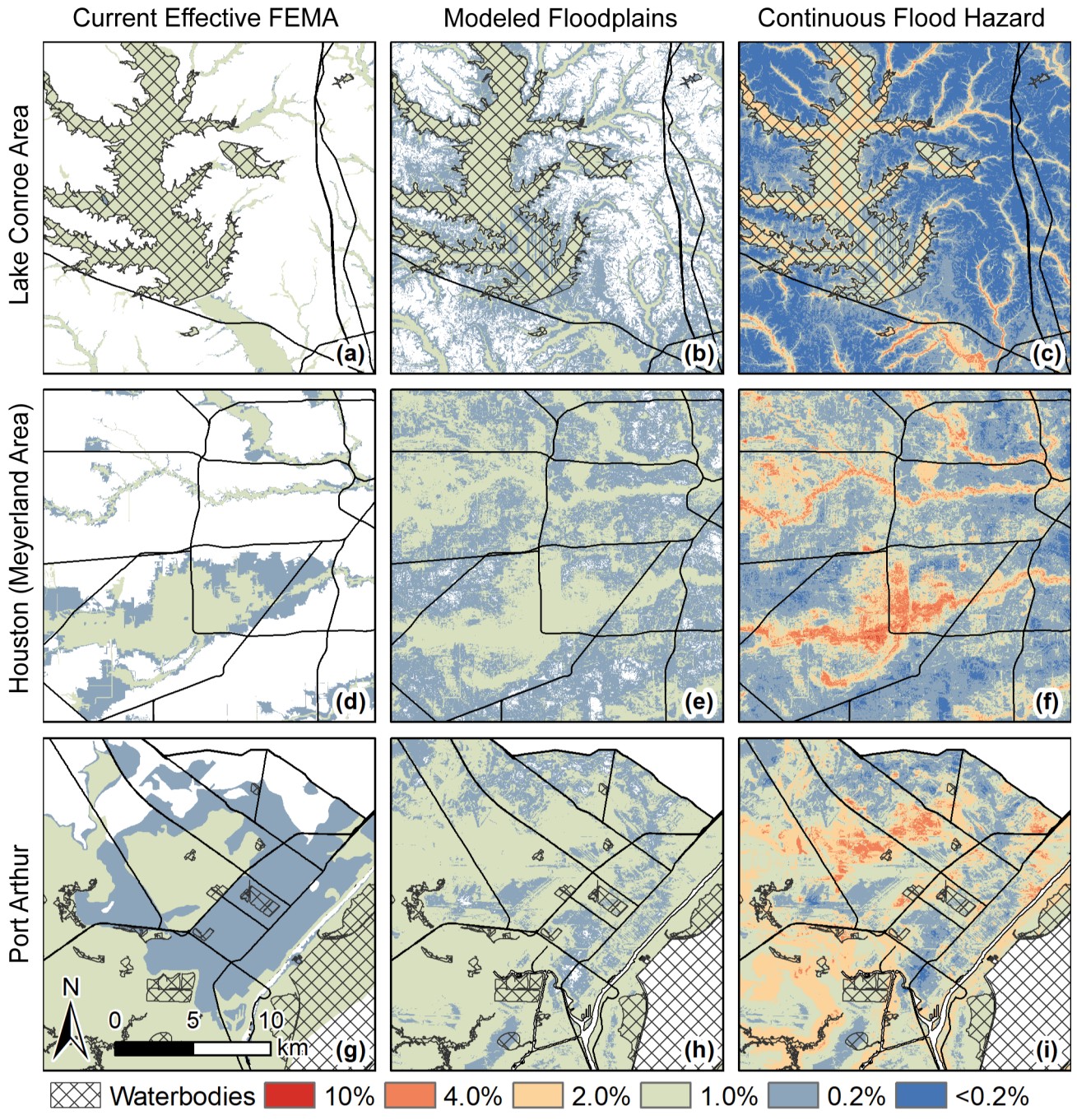

**Figure 7.** Side-by-side visual comparison for three areas affected by differing flood types. (a-c) ~~Lake~~ Conroe ~~area~~ Area (d-f) Houston (Meyerland Area) and (g-i) Port Arthur. Maps compare (left) current effective 100-year and 500-year ~~flood plains~~ FEMA SFHA (c. 2016), (center) areas within the ~~100-year~~ 1% and ~~500-year~~ 0.2% annual flood zones based on the Random Forest ~~flood model;~~ estimate, and (right) continuous flood hazard map. Road layers originated from the U.S. Census Bureau and Floodplains originated from Federal Emergency Management Agency.

**Table 1.** Concept measurements

| Name | Description | Initial Resolution | Range |
|---|---|---|---|
| Accumulated Flow | Hydrologic accumulation for contributing cells | 30 m | $0 - 2.7 * 106$ cells |
| Hydraulic Conductivity (Ksat)[1] | Average soil water transmission for all contributing cells | 30 m | 0-140 $\mu$m/sec |
| Manning's roughness coefficient[2] | Average landcover roughness for all contributing cells | 30 m | $0.001 - 0.39$ |
| Elevation[3] | Digital Elevation Model (DEM) from USGS | 30 m | 0 -155 m |
| Distance to Coast[4] | Euclidean distance from the coast | 10 m | $0 - 155$ km |
| Distance to Stream[4] | Euclidean distance from the nearest ordered stream | 10 m | $0 - 15$ km |
| Height Above Nearest Drainage (HAND)[5] | Relative vertical height compared with nearest stream, lake, or coastline | 10 m | 0- 59 m |
| Imperviousness[6,7] | Percent impervious (NLCD) | 30 m | $0 - 100\%$ |
| Topographic Wetness Index (TWI) | Topographic Wetness Index adjusted to have no zero values | 30 m | $8.35 - 39.53$ |

1 USGS SSURGO Database website https://www.nrcs.usda.gov/

2 Years: 2001, 2004, 2006, 2008, 2011, 2013, 2016

3 Landfire Database website https://www.landfire.gov

4 NHD Plus https://www.usgs.gov/core-science-systems/ngp/national-hydrography/nhdplus-high-resolution

5 Height Above Nearest Drainage https://web.corral.tacc.utexas.edu/nfiedata/

6 NLCD Database website https://www.mrlc.gov/data

7 Years: 2001, 2006, 2011, 2016

**Table 2.** Represents the number of structures with at at least a 1.0% or 0.2% probability of flooding within the modeled Flood Hazard and the SFHA.

| Layer | Probability | Study Area | Lake Conroe | Houston (Meyerland) | Port Arthur |
|---|---|---|---|---|---|
| Flood Hazard | 1.0% | 649,000 | 6,040 | 87,300 | 26,000 |
| | 0.2% | 1,810,000 | 23,100 | 198,000 | 44,300 |
| SFHA | 1.0% | 207,000 | 1,210 | 27,500 | 3,080 |
| | 0.2% | 500,000 | 1,540 | 58,000 | 32,900 |

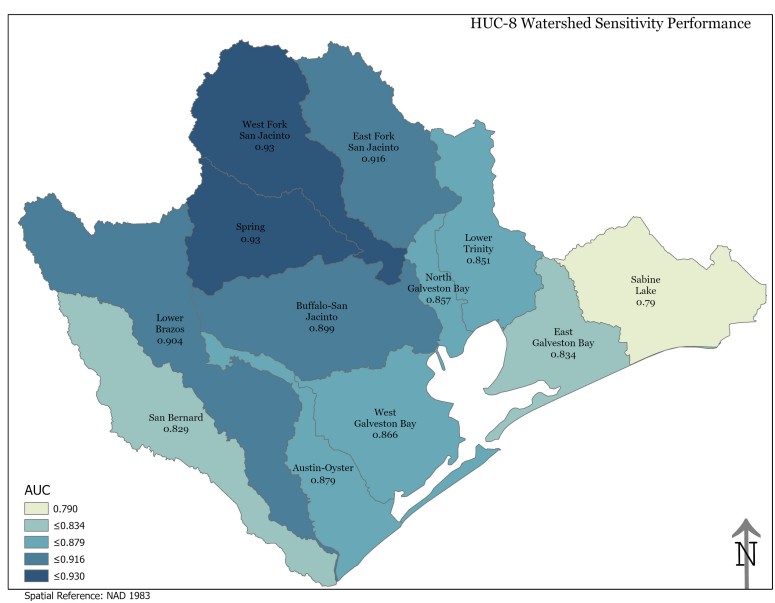

**Figure A1.** ~~Feature importance chart~~Sensitivity performance for each watershed. Note ~~variables below, 0.05 were removed from~~ the ~~final model~~higher elevations occur in the northwest portion of the study area.

**Appendix A**