# Peer review of "Quantification of Continuous Flood Hazard using Random Forrest Classification and Flood Insurance Claims at Large Spatial Scales: A Pilot Study in Southeast Texas"

_Natural Hazards and Earth System Sciences, 2020_

## Referee Comment (RC1) · Anonymous Referee #1 · 23 Nov 2020

General Comments:

This manuscript develops a random forest model using landscape (hydrology, topography) predictors, training the model on flood insurance claims, to develop flood hazard maps. The authors state this is an improvement over previous efforts by including historic records of flood damage to create a probabilistic floodplain. The paper is well written, and the methods are sound. The year-by-year analysis is unique and useful, in that it demonstrates that the model can be updated with new claims (as the authors note in the Conclusion section). It expands on previous applications of floodplain delineation via machine learning, a worthy addition to the field. The case study uses Houston, TX, USA, an area often impacted by flooding.

Specific Comments:

L40: although not ML, physics-based models using the shallow water equation should be mentioned briefly, such as Wing et al. (2017). This was completed for the entire CONUS. Or alternatively, this could be included somewhere in the Background section. Wing, O. E. J., P. D. Bates, C. C. Sampson, A. M. Smith, K. A. Johnson, and T. A. Erickson (2017), Validation of a 30 m resolution flood hazard model of the conterminous United States, Water Resour. Res., 53, 7968–7986, doi:10.1002/ 2017WR020917.

L46: this is a statement that needs more elaboration, why do previous sampling methods undermine model reliability? Why is your proposed method superior (and what is it)?

L118-120: a clarification needed here. The sampling to generate pseudo-absences was based on building footprints that did not have an NFIP claim?

L120-125: How are these one-to-one matched on a parcel that has two buildings, i.e. any combination of a house, shed, and detached garage? How were these situations handled, since the claims were parcel-level, not structure-level?

L205-210: error curve vs. number of trees and/or tree depth would be useful for the appendix.

L210-212: Worth stating the features determined important here.

An additional figure that would be useful to the reader would be to compare a location with the modeled flood for that year versus the parcel-level claims for a single year (or multiple panels for two very different years). Perhaps show a coastal area and an inland area, or contrasting coastal, fluvial, and pluvial? This would complement both fig 5 and fig 2.

[Figure]

L230-235: are water areas masked out from the accuracy assessment? That is, there is no prediction on permanent water bodies? Please clarify.

L267: does this mean that a more nuanced model would perform better? That is, isolating areas based on their characteristics (inland vs. coast) and training/testing separate models to improve performance in both areas, and likely resulting in different important variables?

L296: statistical ML model?

Figure A1: increase font size, difficult to read. In addition, I think this would be worthwhile to include in the main text, given the amount of time devoted to variable discussion in the methods.

Technical Corrections: L39-40: spaces missing here.

L53-55: Are these lines necessary?

---

## Referee Comment (RC2) · Jerom Aerts (Referee) · 17 Dec 2020

General Comments:

The authors developed a novel method for creating continues flood hazard maps using a random forest model trained on a flood insurance claim dataset and independent terrain and hydrological predictors. The necessity of such a method is clearly motivated and shows great potential as a quantitative alternative to the currently in place flood hazard maps. The novelty of the study lies in the use of the insurance claim dataset.

[Figure]

Overall, the manuscript is well written and shows interesting new insights into how random forest models can be used for flood prediction.

Although the majority of this study uses open datasets, the main dataset in question is not open to public due to privacy concerns. Therefore, it is not possible to reproduce the results and to assess the quality and or accuracy of this dataset. An anonymized version of this dataset would elevate the study and benefit the flood modelling community. In addition, the study would benefit from archiving the code used to create the model and analyze the results.

Specific Comments per chapter:

Introduction:

L40-41: The authors propose machine learning models as an alternative. The introduction would benefit from a broader overview of the different types of available flood hazard model. For example, probabilistic catastrophe models or physics-based models. The added value of ML models would then be more clear.

Sampson, C. C., Fewtrell, T. J., O'Loughlin, F., Pappenberger, F., Bates, P. B., Freer, J. E., and Cloke, H. L.: The impact of uncertain precipitation data on insurance loss estimates using a flood catastrophe model, Hydrol. Earth Syst. Sci., 18, 2305–2324, https://doi.org/10.5194/hess-18-2305-2014, 2014.

Wing, O. E. J., P. D. Bates, C. C. Sampson, A. M. Smith, K. A. Johnson, and T. A. Erickson (2017), Validation of a 30 m resolution flood hazard model of the conterminous United States, Water Resour. Res., 53, 7968–7986, doi:10.1002/ 2017WR020917.).

L46-47: This statement needs clarification. What sampling procedures were used by previous studies? How do they undermine model reliability? How is your method better than previous studies? Do previous studies refer to Woznicki et al. (2019) and Knighton et al. (2020)?

Background:

L71: What does "through this process" refer to?

L74-75: Is this the case for all flood hazard models or only for all ML flood hazard models?

Methods and Materials:

L115: Why did the authors choose the period 1976 to 2017? How often is the NFIP dataset updated? Please clarify.

L118-119: In Barbet-Massin et al., 2012 (referenced in this line) one of the conclusions is that pseudo-absences taken too far from the presence data would not be very informative. The spatial extent of the study would impact model performance. Please elaborate on, if and how the methodology constrained the selection of pseudo-random samples being taken to far from the presence data.

L121-125: Due to the importance of this section for the methodology please clarify what is meant by "one-on-one sample by watershed and year" and elaborate on why this reduces potential bias form an unbalanced dataset.

L129: Why and how was the decision made to resample to a 10-meter raster instead of a 30-meter raster? Table1 indicates that most variables are native at a 30-meter resolution.

L153-155, 160-162: The reasoning behind choosing the nearest date due to absent data is valid. However it could have implications (skewed behavior) on the results as the authors stated the importance of land use change. Please highlight this in the discussion chapter.

L202-203: The manuscript could benefit from an overview table or flow chart of method and model setup, e.g. number of trees, tree depth, amount of data points, inputs, outputs.

L212: Figure should be in the main text instead of the appendix. After the introduction

of each variable it should be placed here.

Results:

Are neutral water bodies excluded from the analyses? This would cause a high positive hit rate bias.

L231: As a reader I am interested in the results of the twelve watersheds individually as it gives an indication of spatial model performance. If possible, please add to the appendix and refer in the main text.

L252-258: The text expresses flood probability in percentages while the figure uses return periods.

L253: The text mentions that the plots appear visually similar but in fact differs, a table should be added to clarify the differences.

Discussion:

L260-261: "computationally efficient" although one might expect this from a random forest model there is no mention of model runtime in the manuscript making this statement not well founded.

L261-262: "Overall, the model creates an accurate representation of flood hazard for the study area and demonstrates strong discriminatory power." This conclusion is difficult to make as the SFHA flood hazard maps are deemed outdated. Moreover, the results are not validated against actual flood events.

L267-268: Interesting, please add a range of values, or a plot of slope against model performance.

L272-275: This should be made more clear in the methods chapter.

Conclusions:

L291-293: The study area is sufficient in size that the assumption of homogeneous

precipitation is invalid. Given the importance of mesoscale convective systems for major flood events one can argue that the location and duration of such systems is a main driver for floods.

L300: Although the ease of using TWI is shown in the manuscript it is not used in the final model, please use another alternative driver as an example.

Figure corrections:

Figure 5: This figure could use a clearer description of what is depicted. Headers above each column enhances readability. In addition the figure would benefit from an overview map depicting the location of each column.

Figure A1: Font size is too small. Add a vertical line at 0.05 to indicate which variables are removed. This is only clear based on the text.

Technical corrections:

L28: NFIP abbreviation not written out, first occurrence in L49-50.

L38-39: Missing spaces.

L191: Start new paragraph.

L219: "A calibration plot shows", please refer to the figure number.

L233: "was", typo?

L235: The word "should" seems not necessary in this sentence.

L247-249: Some typesetting errors.

---

## Author Comment (AC1) · 24 Jan 2021

We want to thank the reviewers for their comments and insights. We will address many of their responses by adding to the background section and incorporating figures that we believe will significantly improve the article. Below we have outlined our responses, note many of the comments from both reviewers were similar or dealing with the same sections, because of these similarities we have ordered our responses accordingly.

**R1** General Comments: This manuscript develops a random forest model using landscape (hydrology, topography) predictors, training the model on flood insurance claims, to develop flood hazard maps. The authors state this is an improvement over previous efforts by including historic records of flood damage to create a probabilistic floodplain. The paper is well written, and the methods are sound. The year-by-year analysis is unique and useful, in that it demonstrates that the model can be updated with new claims (as the authors note in the Conclusion section). It expands on previous applications of floodplain delineation via machine learning, a worthy addition to the field. The case study uses Houston, TX, USA, an area often impacted by flooding.

**R2** General Comments: The authors developed a novel method for creating continues flood hazard maps using a random forest model trained on a flood insurance claim dataset and independent terrain and hydrological predictors. The necessity of such a method is clearly motivated and shows great potential as a quantitative alternative to the currently in place flood hazard maps. The novelty of the study lies in the use of the insurance claim dataset.

Overall, the manuscript is well written and shows interesting new insights into how random forest models can be used for flood prediction. Although the majority of this study uses open datasets, the main dataset in question is not open to public due to privacy concerns. Therefore, it is not possible to reproduce the results and to assess the quality and or accuracy of this dataset. An anonymized version of this dataset would elevate the study and benefit the flood modelling community.

In addition, the study would benefit from archiving the code used to create the model and analyze the results. Specific Comments per chapter:

We agree, but are bound by data sharing agreements that do not allow us to share these data due to PII. Unfortunately, this limitation still exists following anonymization since the spatial locations are at the individual level, which is still a violation of PII restrictions. An open-source version of the dataset at a larger spatial unit (census tract) is available at https://www.fema.gov/openfema-data-page/fima-nfip-redacted-claims.

Specific Comments:

**R1**: L40: although not ML, physics-based models using the shallow water equation should be mentioned briefly, such as Wing et al. (2017). This was completed for the entire CONUS. Or alternatively, this could be included somewhere in the Background section.

Wing, O. E. J., P. D. Bates, C. C. Sampson, A. M. Smith, K. A. Johnson, and T. A. Erickson (2017), Validation of a 30 m resolution flood hazard model of the conterminous United States, Water Resour. Res., 53, 7968–7986, doi:10.1002/ 2017WR020917.

**R2**: Introduction: L40-41: The authors propose machine learning models as an alternative. The introduction would benefit from a broader overview of the different types of available flood hazard model. For example, probabilistic catastrophe models or physics-based models. The added value of ML models would then be more clear.

Sampson, C. C., Fewtrell, T. J., O'Loughlin, F., Pappenberger, F., Bates, P. B., Freer, J. E., and Cloke, H. L.: The impact of uncertain precipitation data on insurance loss estimates using a flood catastrophe model, Hydrol. Earth Syst. Sci., 18, 2305–2324, https://doi.org/10.5194/hess-18-2305-2014, 2014.

Wing, O. E. J., P. D. Bates, C. C. Sampson, A. M. Smith, K. A. Johnson, and T. A. Erickson (2017), Validation of a 30 m resolution flood hazard model of the conterminous United States, Water Resour. Res., 53, 7968–7986, doi:10.1002/ 2017WR020917.).

We will add the following paragraph to the background section:

**An alternative approach that is gaining popularity are models that can leverage both big data (e.g., large-scale remotely sensed imagery) and high-performance computing environments to efficiently estimate flood hazard at continental (Bates et al. 2020, Wing et al 2018, Paprotny et al 2017) and global scales (Ward et al 2015, Ikeuchi et al 2017). These models overcome some of the limitations of community-by-community watershed-scale modeling by providing a methodologically consistent approach using comprehensive spatial datasets that is less resource intensive. Although capable of accurately identifying general patterns of flood hazard over large areas, these models often struggle to correctly estimate small scale variability in stormwater dynamics within highly developed areas. That is, model performance at community and household scales is highly dependent on detailed information about local-scale hydrology and flood control infrastructure, with limited observational data to validate the results. Moreover, these models still suffer from the same sources of uncertainty found at the watershed-scale including, specifically, natural uncertainty arising from physical processes driving runoff generation (Singh 1997), uncertainty found within model parameterization (Moradkhani et al. 2008), and uncertainty arising from the quality or availability of input data (Rajib et al. 2020). Accounting for uncertainty at locally-relevant scales would likely require significant increases in observational hydrometeorological data and immense financial resources.**

Moradkhani, H., & Sorooshian, S. (2009). General review of rainfall-runoff modeling: model calibration, data assimilation, and uncertainty analysis. In *Hydrological modelling and the water cycle* (pp. 1-24). Springer, Berlin, Heidelberg.

Rajib, A., Liu, Z., Merwade, V., Tavakoly, A. A., & Follum, M. L. (2020). Towards a large-scale locally relevant flood inundation modeling framework using SWAT and LISFLOOD-FP. *Journal of Hydrology*, *581*, 124406.

Singh, V. P. (1997). Effect of spatial and temporal variability in rainfall and watershed characteristics on stream flow hydrograph. *Hydrological processes*, *11*(12), 1649-1669.

**R1:** L46: this is a statement that needs more elaboration, why do previous sampling methods undermine model reliability? Why is your proposed method superior (and what is it)?

**R2:** L46-47: This statement needs clarification. What sampling procedures were used by previous studies? How do they undermine model reliability? How is your method better than previous studies? Do previous studies refer to Woznicki et al. (2019) and Knighton et al. (2020)?

After further review the statements on sampling in location are not a primary focus of the study and detract from the overall goal. Of estimating a probabilistic floodplain and comparing against the regulatory floodplain. We propose to remove the end of this sentence.

**R2** L71: What does "through this process" refer to?

We propose to change the document here to reduce confusion as follows.

**By modeling flood hazard through this process,** researchers are able to map large geographic areas…

**R2** L74-75: Is this the case for all flood hazard models or only for all ML flood hazard models?

This should be

**Data-driven** flood hazard models.

We will adjust the document accordingly.

Methods and Materials:

**R2** L115: Why did the authors choose the period 1976 to 2017? How often is the NFIP dataset updated? Please clarify.

We will change the sentence as follows:

**NFIP flood claims were used as the predictive variable. NFIP claims between 1976 and 2017 were available for the study totaling 184,826 claims, each of which was geocoded to the parcel centroid.**

**R1** L118-120: a clarification needed here. The sampling to generate pseudo-absences was based on building footprints that did not have an NFIP claim?

**R2** L272-275: This should be made more clear in the methods chapter.

We propose the following changes to the paper to clarify this.

A pseudo-random sample of background values can be used as a proxy for locations where flooding is absent (Barbet-Massin et al., 2012). **When used in this way, the pseudo-absences essentially represent the population that could potentially flood.** Therefore, the background sample locations were based on a random selection of **all** structures **within the study area**. The Microsoft structures footprints dataset was used due to the open source availability and high accuracy

**R2** L118-119: In Barbet-Massin et al., 2012 (referenced in this line) one of the conclusions is that pseudo-absences taken too far from the presence data would not be very informative. The spatial extent of the study would impact model performance. Please elaborate on, if and how the methodology constrained the selection of pseudo-random samples being taken to far from the presence data.

By keeping the background points within the same watershed and year as noted on line 121 we constrain the distance spatially and temporally to minimize this potential bias.

**R1** L120-125: How are these one-to-one matched on a parcel that has two buildings, i.e. any combination of a house, shed, and detached garage? How were these situations handled, since the claims were parcel-level, not structure-level?

Since we are using all structures for our potential background points, we don't account for parcels in this study.

**R2** L121-125: Due to the importance of this section for the methodology please clarify what is meant by "one-on-one sample by watershed and year" and elaborate on why this reduces potential bias form an unbalanced dataset.

We will add the following sentence to reduce confusion:

**Meaning that for each claim, a structure is selected given the same year and located within the same watershed.**

**R2** L129: Why and how was the decision made to resample to a 10-meter raster instead of a 30-meter raster? Table1 indicates that most variables are native at a 30-meter resolution.

We resampled the 30-meter rasters to 10-meter rasters to maintain the information of the rasters that were native at a 10-meter resolution. The raster of particular importance was the Height Above Nearest Drainage (HAND) which can vary significantly over a short distance. Resampling this raster to a higher resolution would have resulted in the loss of critical information such as the use of fill for elevating a parcel footprint.

**R2** L153-155, 160-162: The reasoning behind choosing the nearest date due to absent data is valid. However it could have implications (skewed behavior) on the results as the authors stated the importance of land use change. Please highlight this in the discussion chapter.

We'll identify the limitation within the discussion that predictions of older flood claims may be biased due to temporal variation in the variables.

**Another source of bias comes from how contextual variables were associated with the claims dataset. That is, the contextual variables that were attached to the claims came from the closest year for which the data was made available. However, this can cause some skewed behavior, particularly for the older claims, because some of the contextual variables only start in 2001 whereas the claims date back to 1976. Variables impacted the most by this are the result of large-scale changes in urbanization over time, which include imperviousness and roughness. More specifically, the imperviousness and roughness conditions attached to claims is going be less representative of the actual conditions during the time of loss the further one goes back in time. However, a large majority of the claims (approximately 75%) occurred after 2001 and most of the contextual variables do not change, or change minimally, over time (e.g. HAND, elevation, distance to coast, etc.) mitigating the influence of this bias.**

**R2** L202-203: The manuscript could benefit from an overview table or flow chart of method and model setup, e.g. number of trees, tree depth, amount of data points, inputs, outputs.

This is a great comment that should help to reduce reader confusion we propose to include the following flow chart.

[Figure]

**R2** L205-210: error curve vs. number of trees and/or tree depth would be useful for the appendix.

Error curves regarding number of trees has been extensively explored within the literature. Ours falls within normal curves (see Liaw et al), and we don't feel it really add's to the studies narrative.

Liaw, A., & Wiener, M. (2002). Classification and regression by randomForest. *R news*, *2*(3), 18-22.

**R2** L212: Figure should be in the main text instead of the appendix. After the introduction of each variable it should be placed here.

**R1** Figure A1: increase font size, difficult to read. In addition, I think this would be worthwhile to include in the main text, given the amount of time devoted to variable discussion in the methods.

**R2** Figure A1: Font size is too small. Add a vertical line at 0.05 to indicate which variables are removed. This is only clear based on the text.

These great recommendations and we will incorporate this change like below.

[Figure]

**R2** L210-212: Worth stating the features determined important here.

We propose to change the text as follows.

Initially, all variables were added to the model, those variables with the lowest contribution to feature importance were removed from the final model. **Two variables were removed from the final model for not meeting the required threshold TWI and Flow Accumulation** (Figure ).

**R2** An additional figure that would be useful to the reader would be to compare a location with the modeled flood for that year versus the parcel-level claims for a single year (or multiple panels for two very different years). Perhaps show a coastal area and an inland area, or contrasting coastal, fluvial, and pluvial? This would complement both fig 5 and fig 2.

The goal of the article is to outline the process for estimating flood hazard across multiple years, while we provide some insight into how the model performs year to year. We don't feel it's in the scope of the article to show individual year flood hazard as that may detract from the long-term estimates and ultimately confuse the reader.

**R1** L230-235: are water areas masked out from the accuracy assessment? That is, there is no prediction on permanent water bodies? Please clarify.

**R2** Are neutral water bodies excluded from the analyses? This would cause a high positive hit rate bias.

This is a viable error potential with the standard process of using background points randomly selected across the entire study are. However, we used structures for a stratified sample, which are typically not located within waterbodies, and therefore these areas are not included within the accuracy assessments.

**R2** L231: As a reader I am interested in the results of the twelve watersheds individually as it gives an indication of spatial model performance. If possible, please add to the appendix and refer in the main text.

We will include this figure within the appendix.

**R2** L252-258: The text expresses flood probability in percentages while the figure uses return periods.

We will change the figure accordingly.

**R2** L253: The text mentions that the plots appear visually similar but in fact differs, a table should be added to clarify the differences.

We will add a table to help reduce confusion.

Discussion:

**R2** L260-261: "computationally efficient" although one might expect this from a random forest model there is no mention of model runtime in the manuscript making this statement not well founded.

The model is computationally efficient, scalable, and can be used to predict flood hazards over relatively large regions. **Training and image prediction was run in 1-hr on a high end desktop computer, in comparison physical models can run for 20 minutes or up to 10 hours for an area 0.15% of the study area** (Apel, Aronica, Kreibich, & Thieken, 2009)**.** Overall, the model creates an accurate representation of …

Apel, H., Aronica, G., Kreibich, H. & Thieken, A. (2009). Flood risk analyses—how detailed do we need to be? *Natural Hazards*, *49*(1), 79–98.

**R1** L267: does this mean that a more nuanced model would perform better? That is, isolating areas based on their characteristics (inland vs. coast) and training/testing separate models to improve performance in both areas, and likely resulting in different important variables?

A more nuanced model likely would perform better, but based on the data available, the model can't really be nuanced in that way.

**R2** L267-268: Interesting, please add a range of values, or a plot of slope against model performance.

We can add in the image of sensitivities and discuss slope changes along the coast and inland.

**R2** L261-262: "Overall, the model creates an accurate representation of flood hazard for the study area and demonstrates strong discriminatory power." This conclusion is difficult to make as the SFHA flood hazard maps are deemed outdated. Moreover, the results are not validated against actual flood events.

The accurate representation is based on the comparison of NFIP flood claims and at this point in the discussion not comparing with the SFHA. We will rephrase the sentence to reduce confusion.

**Overall, the model demonstrates strong predictive power for estimating historic damages and accurately represents the spatial distribution of the flood hazard across those 40-years of flood damages.**

Conclusions:

**R2** L291-293: The study area is sufficient in size that the assumption of precipitation is invalid. Given the importance of mesoscale convective systems for major flood events one can argue that the location and duration of such systems is a main driver for floods.

We'll update the sentence as follows:

**The model does not currently incorporate precipitation patterns. Future work should examine the sensitivity of the model to precipitation as an input.**

**R1** L296: statistical ML model?

Yes, we will correct this.

**R2** L300: Although the ease of using TWI is shown in the manuscript it is not used in the final model, please use another alternative driver as an example.

We will remove the mention of TWI here.

Figure corrections:

**R2** Figure 5: This figure could use a clearer description of what is depicted. Headers above each column enhances readability. In addition the figure would benefit from an overview map depicting the location of each column.

We will update this.

Technical corrections:

We will fix the technical corrections.

 L28: NFIP abbreviation not written out, first occurrence in L49-50.

L38-39: Missing spaces.

L191: Start new paragraph.

L219: "A calibration plot shows", please refer to the figure number.

 L233: "was", typo?

 L235: The word "should" seems not necessary in this sentence.

 L247-249: Some typesetting errors.

L39-40: spaces missing here.

 L53-55: Are these lines necessary?

We believe these lines are necessary as they help to outline our article to the reader which should help them follow along.